theoretical biology, microbiology, ecology

abundant resource premium, catabolism, mutualism, microbial consortium, mathematical model, microbial ecology

**Author for correspondence:**
Mayumi Seto
e-mail: seto@ics.nara-wu.ac.jp

# Microbial material cycling, energetic constraints and ecosystem expansion in subsurface ecosystems

Mayumi Seto[1] and Yoh Iwasa[2]

[1]Department of Chemistry, Biology, and Environmental Sciences, Nara Women's University, Kita-Uoya Nishimachi, Nara 630-8506, Japan
[2]Department of Bioscience, School of Science and Technology, Kwansei Gakuin University, Gakuen 2-1, Sanda-shi, Hyogo 669-1337, Japan

MS, 0000-0003-4709-5206

To harvest energy from chemical reactions, microbes engage in diverse catabolic interactions that drive material cycles in the environment. Here, we consider a simple mathematical model for cycling reactions between alternative forms of an element (A and $A_e$), where reaction 1 converts A to $A_e$ and reaction 2 converts $A_e$ to A. There are two types of microbes: type 1 microbes harness reaction 1, and type 2 microbes harness reaction 2. Each type receives its own catabolic resources from the other type and provides the other type with the by-products as the catabolic resources. Analyses of the model show that each type increases its steady-state abundance in the presence of the other type. The flux of material flow becomes faster in the presence of microbes. By coupling two catabolic reactions, types 1 and 2 can also expand their realized niches through the abundant resource premium, the effect of relative quantities of products and reactants on the available chemical energy, which is especially important for microbes under strong energetic limitations. The plausibility of mutually beneficial interactions is controlled by the available chemical energy (Gibbs energy) of the system. We conclude that mutualistic catabolic interactions can be an important factor that enables microbes in subsurface ecosystems to increase ecosystem productivity and expand the ecosystem.

## 1. Introduction

Biogeochemical cycles are networks of chemical reactions or physical processes that are driven by organisms. In terrestrial ecosystems, the carbon cycle between carbon dioxide and organic matter under aerobic conditions is predominantly accomplished by plants and decomposers (or aerobic respiratory organisms) by coupling the catabolic pathways to synthesize ATP and anabolic pathways to build biomass by consuming ATP (figure 1a). Aerobic respiration is one of the catabolic reactions, that is, the chemical reactions functioning as energy sources to synthesize ATP. Most catabolic reactions are classified as oxidation–reduction (redox) reactions. They are characterized by electron transfer from an electron-donor compound (organic matter for aerobic respiration) to an electron-acceptor compound (oxygen for aerobic respiration) (figure 1a). Electron transfer establishes an electrochemical gradient on the cell membrane, which eventually powers ATP synthesis (more specifically ATP synthesis by the oxidative phosphorylation) [1,2].

We focus here on the material cycles between two alternative forms of an element, which can be achieved only through catabolic reactions using chemical reactions as direct energy sources. For instance, microbial catabolism is considered the dominant driver of the iron cycle between Fe(II) (iron in its +2 oxidation state) and Fe(III) (iron in its +3 oxidation state) in most

**Figure 1.** Examples of the simplest element cycle between two states of an element. (*a*) Carbon cycle between carbon dioxide and organic matter (*nota bene*, $CH_2O$ in figure 1 is a general stoichiometric representation of plant-derived organic matter). (*b*) Iron cycle between Fe(II) (ferrous iron) and Fe(III) (ferric iron). Fe(II) can be $Fe^{2+}$, Fe(II) complex or Fe(II) oxides/hydroxides, and Fe(III) can be $Fe^{3+}$, Fe(III) complex or Fe(III) oxides/ hydroxides. (*c*) Carbon cycle between carbon dioxide and methane. $e^-$ denotes an electron.

environments [3–6]. The iron cycle driven by two possible catabolic reactions is illustrated in figure 1*b*. Some bacteria harvest energy from iron oxidation by coupling Fe(II) and oxygen (or an alternative electron-acceptor compound), in which electrons are transferred from Fe(II) to oxygen and then Fe(III) is produced. Other bacteria harness iron reduction, which transforms Fe(III) into Fe(II) by adding electrons that are obtained from an electron-donor compound. Iron-oxidizing and iron-reducing bacteria are likely to be separated spatially and temporally, but some researchers suggest microscale cycling of iron occurring with iron-oxidizing and iron-reducing bacteria in close proximity to each other [7–9]. Methanogenic archaea and methane-oxidizing bacteria may also drive the carbon cycle between carbon dioxide and methane (figure 1*c*). Although microbial methane production and consumption account for more than half of the estimated global methane production and consumption [10,11], the underlying microbial processes remain unclear. Some methanogenic archaea produce methane by transferring electrons from hydrogen gas to carbon dioxide. Methane-consuming microbes obtain energy by transferring electrons from methane to oxygen gas, sulfate, nitrate, iron and manganese oxides, which eventually produce carbon dioxide (or hydrogen carbonate) as a by-product [12–15]. The association of these two microbes may cycle methane within the seafloor zones [16].

Two distinct catabolic types in a microbial consortium (assemblage) involved in a microscale cycle should be mutualistic because they feed each other with a compound that is a by-product for one but a catabolic resource for the other. These two types are also beneficial to each other in terms of the amount of energy that they can generate from catabolic reactions. According to thermodynamics, the negative of the Gibbs energy change of a reaction ($-\Delta_r G$ in kJ mol$^{-1}$) provides the maximum available energy per unit of reaction progress (i.e. when one mole of a catabolic reaction occurs).

$-\Delta_r G$ of a reaction $A + B \rightarrow C + D$ is defined as

$$- \Delta_r G = -\Delta_r G° + RT \ln \frac{\{A\}\{B\}}{\{C\}\{D\}}, \tag{1.1}$$

where $-\Delta_r G°$ indicates the negative change in the standard Gibbs energy of the reaction (in kJ mol$^{-1}$), which will be explained in §2, $R$ is the gas constant ($R = 8.13 \times 10^{-3}$ kJ mol$^{-1}$ K$^{-1}$) and $T$ is the absolute temperature (in K). Symbol $\{X\}$ represents the activity of X, calculated as $\{X\} = a_x[X]$, where $a_x$ is the activity coefficient of X and $[X]$ is its molar concentration. The second term on the right-hand side of equation (1.1) is called the 'abundant resource premium' (ARP) [17,18]. This quantity indicates the change in the available energy, which is increased by the abundance of catabolic resources and decreased by the abundance of by-products. Since the microbial catabolic interactions involved in the element cycle between two states increase the ARP, they may enhance the productivity of both catabolic reactions. Because many microbes observed in subsurface environments harness reactions with small values of $-\Delta_r G°$, their survival may depend on the significance of ARP. The mutualistic interaction via ARP has also been observed between archaea and bacteria, which is mediated by interspecies $H_2$ transfer, which might lead the origin of eukaryotic cell [19].

In a previous study, we demonstrated that microbes of one type (e.g. type 2) that use a catabolic by-product of the other type (type 1) can increase the fitness of type 1 because the type 2 microbe consumes the by-product of the type 1 microbe and enhances the ARP of the energy-generating reaction for type 1 [18]. In the present paper, we analyse the case in which the by-products of each of the two types are among the catabolic resources of the other type because an element is recycled by the two types of microbe, as illustrated in figure 2. We show that the biomass of each type is enhanced by the presence of the other, and the material flux between alternative forms becomes faster

**Figure 2.** Flow diagram of the model.

in the presence of both types. In addition, in a system of coupled catabolic reactions, the presence of the ARP term expands the geochemical conditions in which types 1 and 2 survive.

## 2. Model

We consider two types of microbe, referred to as types 1 and 2, each of which harnesses a different redox reaction, referred to as reactions 1 and 2, between the two compounds (A and $A_e$) comprising the same element, as illustrated in figure 2. Type 1 or 2 may consist of multiple strains, rather than a single species, but microbes of the same type should share the same energy source and the same redox reaction. Types 1 and 2 build a consortium (a flock of cells) in which two types separately form clusters of the same type. This causes a strong intraspecific interaction as they are in the close vicinity. Nevertheless, the clusters of types 1 and 2 are within a distance sufficient to exchange their catabolic by-products.

Type 1 generates energy from the following overall reaction:

$$A + e^- \rightarrow A_e \quad (A \text{ reduction}), \qquad (2.1a)$$
$$B_e \rightarrow B + e^- \quad (B \text{ oxidation}) \qquad (2.1b)$$
and $\quad A + B_e \rightarrow A_e + B$

(reaction 1: the overall reaction of equations (2a) and (2b)),
$$\qquad (2.1c)$$

where $e^-$ indicates an electron, and X and $X_e$ are two different compounds, or two different redox states, composed of the same element. $X_e$ has more electrons than X. Most catabolic reactions are accompanied by $H_2O$, $H^+$ and $OH^-$ to balance the stoichiometry, but we consider here the situation in which the molar concentrations of $H^+$ and $OH^-$ are constant because of the buffering of the medium.

The type 2 microbe generates energy by coupling the reverse reaction of equation (2.1a) with the other electron

acceptor compound, C:

$$A_e \rightarrow A + e^- \quad (A \text{ oxidation}), \qquad (2.2a)$$
$$C + e^- \rightarrow C_e \quad (C \text{ reduction}), \qquad (2.2b)$$
and $\quad A_e + C \rightarrow A + C_e$

(reaction 2: the overall reaction of equations (2.2a) and (2.2b)).
$$\qquad (2.2c)$$

According to the custom, the molar concentrations of compounds A, B and C are denoted by italic letters in the following.

We assume that an element comprising compounds A and $A_e$ is completely recycled on the microscale, so that the total amount of this element in the vicinity of type 1 and 2 microbes is conserved ($A_T = A_e + A$). Furthermore, to build a simple, mathematically tractable model, all of the activity coefficients are assumed to be unity, and the concentrations of all other compounds ($B$, $B_e$, $C$ and $C_e$) are constant because of the existence of buffering capacity. Let $x_1$ and $x_2$ be the biomass contents of types 1 and 2, respectively. The dynamics of $x_1$, $x_2$ and $A$ are given as follows:

$$\frac{dA}{dt} = -k_1 A B_e + k_2 (A_T - A)C - r_1 \frac{A}{K_1 + A}\frac{B_e}{K_{1B_e} + B_e}x_1$$
$$+ r_2 \frac{(A_T - A)}{K_2 + (A_T - A)}\frac{C}{K_{2C} + C}x_2, \qquad (2.3a)$$
$$\frac{dx_1}{dt} = (F_1(A) - m_1 - s_1 x_1)x_1 \qquad (2.3b)$$
and $\quad \dfrac{dx_2}{dt} = (F_2(A) - m_2 - s_2 x_2)x_2, \qquad (2.3c)$

where $k_i$ is the abiotic reaction rate constant of reaction $i$, $r_i$ is the maximum catalytic rate per unit of biomass and $K_i$ is the Michaelis–Menten coefficient for compound A used by reaction $i$. The first and second terms of equation (2.3a) are the abiotic reaction rates of reactions 1 and 2, which are typically proportional to the product of the amounts of the reactants ($A$ and $B_e$ for reaction 1, and $A_e$ and $C$ for reaction 2), following the law of mass action, with the rate constant $k_i$. The third and fourth terms of equation (2.3a) are the microbial reaction rates of reactions 1 and 2, respectively, which follow the dual-limitation kinetics of the Monod equation (i.e. the Michaelis–Menten equation) because both the abundance of electron donor and acceptor can limit the reaction rate [20].

$F_i(A)$ is the catabolic-based growth rate of type $i$, $m_i$ is the maintenance energy loss rate of type $i$ and $s_i$ is the density-dependent mortality rate constant of type $i$. The biomass of type $i$ decreases because of energy dissipation for maintenance and the local self-regulating detrimental effects of high population density, such as the competition for physical space, because cells of the same catabolic type are often densely colonized in a microhabitat rather than freely residing in a medium.

We consider the model targeting microorganisms, such as microbes inhabiting the deep subsurface, the growth of which is limited by the energy that they can obtain from their catabolic reactions [17]. The maximum energy production rate when type $i$ catalyses reaction $i$ is calculated as the microbial catalytic rate multiplied by the negative of the Gibbs energy change of reaction $i$. The catabolic-based growth rates of types 1 and 2 are defined as follows:

$$F_1(A) = q_1 c_1 r_1 \frac{A}{K_1 + A}\frac{B_e}{K_{1B_e} + B_e}(-\Delta_r G_1), \qquad (2.4a)$$

and

$$F_2(A) = q_2 c_2 r_2 \frac{A_T - A}{K_2 + (A_T - A)} \frac{C}{K_{2C} + C}(-\Delta_r G_2), \qquad (2.4b)$$

where

$$-\Delta_r G_1 = -\Delta_r G_1^\circ + RT\ln\frac{A B_e}{(A_T - A)B} \qquad (2.4c)$$

and

$$-\Delta_r G_2 = -\Delta_r G_2^\circ + RT\ln\frac{(A_T - A)C}{A C_e}. \qquad (2.4d)$$

In equations (2.4a) and (2.4b), $q_i$ is the biomass yield of species $i$ for a given energy gain from ATP that is generated from a catabolic reaction, $c_i$ is the fraction of energy that can be used for ATP synthesis ($0 < c_i < 1$), excluding energy expenditure such as loss by heat transfer, and $-\Delta_r G_i$ corresponds to $-\Delta_r G$ of reaction $i$ as described by equation (1.1). The second terms on the right-hand side of equations (2.4c) and (2.4d) are the ARPs of reactions 1 and 2.

$-\Delta_r G_i^\circ$ is the standard Gibbs energy change of reaction $i$, where $^\circ$ indicates the standard state condition in which all reactants and products have activity $= 1$. $-\Delta_r G^\circ$ is the difference between the sum of the Gibbs energy change of formation, $\Delta_f G^\circ$, for the reactants and the same sum for the by-products:

$$-\Delta_r G_1^\circ = \Delta_f G_A^\circ + \Delta_f G_{B_e}^\circ - (\Delta_f G_{A_e}^\circ + \Delta_f G_B^\circ) \qquad (2.4e)$$

and

$$-\Delta_r G_2^\circ = \Delta_f G_{A_e}^\circ + \Delta_f G_C^\circ - (\Delta_f G_A^\circ + \Delta_f G_{C_e}^\circ), \qquad (2.4f)$$

where $\Delta_f G_x^\circ$ indicates the Gibbs energy change of $x$ (here, $x =$ A, $A_e$, B, $B_e$, C or $C_e$) formation and is the relative level of Gibbs energy of $x$ from all of the reference states of the elements composing $x$ (e.g. the relative level of Gibbs energy of $CH_4$ in comparison with C(s, graphite) and $H_2$(g)). Hence, the value of $\Delta_f G_x^\circ$ is intrinsic to $x$, and $-\Delta_r G^\circ$ is a reaction-specific quantity in the standard condition. A low value of $\Delta_f G_x^\circ$ indicates that compound $x$ is stable, and more energy can be generated from a redox reaction that converts reactants with a higher $\Delta_f G_x^\circ$ to compounds with a lower $\Delta_f G_x^\circ$.

# 3. Results

## (a) Steady states

We explain all of the mathematical analyses in the appendix (see electronic supplementary material). In the following, we describe the results only. The behaviour of the model can be understood very clearly. The model has three variables, namely, $A$, $x_1$ and $x_2$. There exists a single locally stable steady state. This steady state is globally stable according to the numerical analyses of the model starting from different initial conditions. Hence, we focus on the nature of the stable steady state in the following.

We introduce two functions of $A$: $\varphi_1(A)$ and $\varphi_2(A)$, defined as

$$\varphi_1(A) = \begin{cases} \frac{1}{s_1}(F_1(A) - m_1), & \text{if it is positive} \\ 0, & \text{otherwise} \end{cases} \qquad (3.1a)$$

and

$$\varphi_2(A) = \begin{cases} \frac{1}{s_2}(F_2(A) - m_2), & \text{if it is positive} \\ 0, & \text{otherwise} \end{cases}. \qquad (3.1b)$$

By setting equations (2.3b) and (2.3c) equal to zero, the biomass contents of the two types $x_1$ and $x_2$ at the stable steady state are given by $x_1 = \varphi_1(A)$ and $x_2 = \varphi_2(A)$, respectively. We can represent these two values as functions of $A$, as shown in the two curves in the lower panel of figure 3. Once we know the value of $A$ at the steady state, equation (2.3) gives the values of $x_1$ and $x_2$, as illustrated in the graphs. We can see that $x_1 > 0$ and $x_2 > 0$ in the middle portion of the horizontal axis, labelled as $A_1 < A < A_2$. Outside of this interval, one of the two types is non-existent: $x_1 = 0$ and $x_2 > 0$ for $0 < A < A_1$, and $x_1 > 0$ and $x_2 = 0$ for $A_2 < A < A_T$ ($A_T = 1$). If the steady-state value of $A$ is within one of these three intervals, $x_1$ and $x_2$ are both positive, or one of them is zero.

To determine the value of $A$ at the steady state, we must use equation (2.3a), which indicates the steady-state value of $A$ determined by the balance of two forces.

$$\frac{dA}{dt} = -\psi_1(A, \varphi_1(A)) + \psi_2(A, \varphi_2(A)), \qquad (3.2a)$$

where

$$\psi_1(A, x_1) = A\left[k_1 B_e + r_1 \frac{1}{K_1 + A} \frac{B_e}{K_{1B_e} + B_e} x_1\right] \qquad (3.2b)$$

and

$$\psi_2(A, x_2) = (A_T - A)\left[k_2 C + r_2 \frac{1}{K_2 + (A_T - A)} \frac{C}{K_{2C} + C} x_2\right]. \qquad (3.2c)$$

Intuitively speaking, $\psi_1(A) = \psi_1(A, \varphi_1(A))$ is the rate at which A is converted to $A_e$ by abiotic and microbial reaction 1, and $\psi_2(A) = \psi_2(A, \varphi_2(A))$ is the rate at which $A_e$ is converted to A by abiotic and microbial reaction 2. The upper panels of figure 3 illustrate the rates of these two processes. The steady-state value of $A$ is determined from the balance between these two processes. The broken line with a positive slope indicates $k_1 B_e A$, which is the rate of reaction 1 performed abiotically. For $0 < A < A_1$, the type 1 microbe is absent, and hence, $\psi_1(A)$ and $k_1 B_e A$ are equal. For $A > A_1$, the type 1 microbe is present, and hence, the curve $\psi_1(A)$ is above the broken line because of the enzymatic activity of the type 1 microbe. In a similar manner, the type 2 microbe is present for $A < A_2$, and the curve $\psi_2(A)$ is above the broken line with a negative slope, $k_2 C(A_T - A)$, indicating the rate of reaction 2 performed abiotically.

In figure 3a, the intersection of curve $\psi_1(A)$ and curve $\psi_2(A)$ is labelled as $\hat{R}$. This indicates the steady state with both microbes present, where the abundance of A is indicated as $\hat{A}$. By contrast, the intersection of curve $\psi_1(A)$ and the broken line with a negative slope is labelled as $R_1$. This indicates the steady state with only the type 1 microbe present, and the steady-state abundance of $A$ is $\overline{\overline{A}}_1$. By contrast, the intersection of the two broken lines (one with a positive slope and the other with a negative slope) corresponds to the steady state with neither microbe present, labelled as $R_0$. The value of $A$ at $R_0$ is $A_0$. We can clearly see that $\overline{\overline{A}}_1 < A_0$, indicating that the abundance of A is reduced because of the consumption by the type 1 microbe. We also note that $\overline{\overline{A}}_1 < \hat{A}$ indicates that

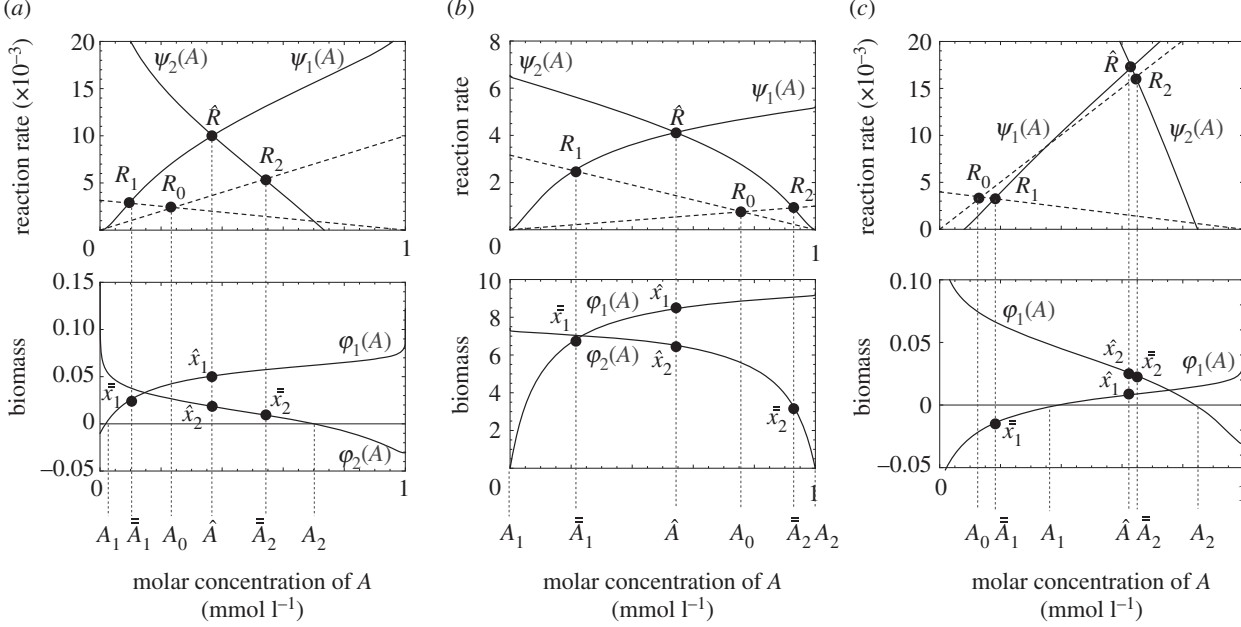

**Figure 3.** The reaction rate from A to $A_e$ and that from $A_e$ to A (upper panels) and the biomass contents of two microbes (lower panels) for the cases where the coexisting steady state $(\hat{A}, \hat{x}_1, \hat{x}_2)$ is the only locally stable steady state, while each type 1 or 2 can solely invade the system for (a) and (b) and while only type 2 can solely invade the system for (c). The only difference between (a) and (b) is the values of $-\Delta_r G_1°$ and $-\Delta_r G_2°$, which highlights the effect of relative significance of ARP (high for (a) and low for (b)) on the steady state. The horizontal axis shows the molar concentration of A. The two solid curves in the upper panels are $\psi_1(A) = \psi_1(A, \varphi_1(A))$ and $\psi_2(A) = \psi_2(A, \varphi_2(A))$, defined as equations (3.2b) and (3.2c), indicating the rate of reaction from A to $A_e$ and the rate of reaction from $A_e$ to A, respectively. The two broken lines are $k_1 A B_e$ and $k_2(A_T − A)C$, indicating the rates of two abiotic reactions. The two curves in the lower panels are $\varphi_1(A)$ and $\varphi_2(A)$, defined as equations (3.1a) and (3.1b). There are possible steady states: neither type 1 nor 2 exists at $(A_0, 0, 0)$; type 1 only exists at $(\overline{\overline{A}}_1, \overline{\overline{x}}_1, 0)$; type 2 only exists at $(\overline{\overline{A}}_2, 0, \overline{\overline{x}}_2)$; both types 1 and 2 exist at $(\hat{A}, \hat{x}_1, \hat{x}_2)$. $R_0$, $R_1$, $R_2$, and $\hat{R}$ correspond to the steady-state reaction rate from A to $A_e$ when neither type 1 nor 2 exists, type 1 only exists, type 2 only exists, and both types exist, respectively. The parameter values of $(m_1, k_1, k_2, r_1, r_2, -\Delta_r G_1°$ and $-\Delta_r G_2°)$ are $(0.1, 10^{-2}, 10^{-2.5}, 10^{-0.8}, 0.5, 50$ and $10)$ for (a), $(0.1, 10^{-2}, 10^{-2.5}, 10^{-0.8}, 0.5, 2000$ and $1600)$ for (b) and $(0.4, 10^{-1.6}, 10^{-2.4}, 10^{-0.9}, 0.5, 50$ and $10)$ for (c). Other parameters are set to the default values presented in electronic supplementary material, table S1.

the abundance of A is enhanced by the presence of the type 2 microbe, which produces A from $A_e$.

## (b) Enhancement of biomass and cycling by the presence of partner type

By noting the value of A obtained from the upper panels of figure 3, we can obtain the steady-state value of the type 1 microbe using the curve of $x_1 = \varphi_1(A)$ in the lower panels. It is $\overline{\overline{x}}_1 = \varphi_1(\overline{\overline{A}})$ and $\hat{x}_1 = \varphi_1(\hat{A})$. We note that $\hat{x}_1$ is greater than $\overline{\overline{x}}_1$ because function $\varphi_1(A)$ is monotonically increasing for the portion it is positive. This implies that the abundance of the type 1 microbe is enhanced by the presence of the type 2 microbe. Not only for the cases illustrated in figure 3, we can conclude that, in general, the abundance of each type of microbe is enhanced by the presence of the other type $(\hat{x}_1 > \overline{\overline{x}}_1 > 0$ and $\hat{x}_2 > \overline{\overline{x}}_2 > 0)$, regardless of the significance of the ARP (figure 3b). This can be proved mathematically for general cases. The proof is given in the electronic supplementary material, appendix. Similarly, the height of point $\hat{R}$ is greater than that of point $R_1$, which is greater than that of point $R_0$. This implies that the rate of cycling is faster in the presence of both types than in systems with only one type, regardless of the significance of the ARP (compare with upper panels in figure 3a,b).

## (c) Mutualistic niche overlapping and ARP-driven niche

Figure 3c indicates that the abundance of A in the absence of both types of microbe is $A_0$, which is smaller than $A_1$. Hence, the type 1 microbe cannot invade the system. However, the

type 2 microbe can invade the system and make $\overline{\overline{A}}_2$ much larger than $A_0$ as the type 2 microbe produces A. Then, the type 1 microbe can invade the system with the type 2 microbe and make the coexisting steady state with the abundance of $\hat{A}$. Hence, the type 1 microbe can exist in the system only with its mutualistic partner (type 2), which indicates that type 2 can construct type 1's niche.

Here, we consider the niche spaces of types 1 and 2 in the $(B, C_e)$ plane. It should be stressed that the growth of types 1 and 2 is insensitive to the abundance of $B$ and $C_e$ if the ARP terms do not exist. The ARP is responsible for the growth responses observed in the $(B, C_e)$ plane. In the absence of microbes, the steady-state concentration of A is

$$A_0 = \frac{A_T k_2 C}{k_1 B_e + k_2 C}. \tag{3.3a}$$

The steady states $-\Delta_r G$ of reactions 1 and 2 in the absence of microbes (denoted by $-\Delta_r G_{10}$ and $-\Delta_r G_{20}$, respectively) are obtained by substituting equation (3.3a) into equations (2.4c) and (2.4d):

$$-\Delta_r G_{10} = -\Delta_r G_1° + RT\ln\frac{k_2 C}{k_1 B} \tag{3.3b}$$

and

$$-\Delta_r G_{20} = -\Delta_r G_2° + RT\ln\frac{k_1 B_e}{k_2 C_e}, \tag{3.3c}$$

where $-\Delta_r G_{10}$ and $-\Delta_r G_{20}$ can be considered the abiotic energetic constraints that are purely determined by physico-chemical processes. According to thermodynamics, the type 1 (or 2) microbe

can exist in the system if $-\Delta_r G_{10}$ (or $-\Delta_r G_{20}$) of the system is large enough to support microbial growth (figure 4a). Hence, from the ecological viewpoint, a system with relatively large $-\Delta_r G_{10}$ (or $-\Delta_r G_{20}$) is a vacant niche potentially available for type 1 (or 2). Although the available niche spaces for types 1 and 2 may overlap on the $(B, C_e)$ plane, there is no conflict between types 1 and 2 as they do not compete for the same resource.

When a small number of type 1 (or 2) microbes are added to the steady state without microbes, the niche of type 1 (or 2) is constrained by the amount of B (or $C_e$) or the value of $-\Delta_r G_{10}$ (or $-\Delta_r G_{20}$) (figure 4a,b). The prevalence of type 1 (or 2) can expand the niche space of type 2 (or 1) in the $(B, C_e)$ plane (figure 4c,d). We adopted the following numerical methods to evaluate the sizes of the fundamental and realized niches of these microbes and to evaluate how they are changed by the presence of the ARP. On a plane where both axes are $\ln B$ and $\ln C_e$, we considered a parameter region of a square shape: $-3 < \ln B < 1$ and $-3 < \ln C_e < 1$. Then, we separated the squared region into $N_f \times N_f$ small squares ($N_f = 51$) of equal size ($4/51 \times 4/51$). We regarded them as a vacant system that can be occupied by microbes, and we counted the number of the squared regions invaded by microbes. If type 1 (or 2) alone can invade the system with B and $C_e$, the point $(B, C_e)$ is referred to as the fundamental niche of type 1 (or 2). In a similar manner, if type 1 (or 2) can invade the system in association with its mutualistic partner, the point $(B, C_e)$ is referred to as the realized niche of type 1 (or 2). The size of the realized niche or the fundamental niche was measured by the number of small squares with these properties.

The increase in the maximum catalytic rate and the decrease in the Michaelis–Menten constant for A (or $A_e$) of type 1 (or 2) increase both the fundamental and realized niches of type 1 (or 2) and the realized niche of its mutualistic partner (figure 5a,b). This suggests that type 1 (or 2) with stronger catalytic ability (higher $r$ or lower $K$) has a larger potential for niche expansion of itself and type 2 (or 1). The presence of type 2 (or 1) effectively enlarges the realized niche of type 1 (or 2), especially when the abiotic reaction rate constant of reaction 2 (or 1) is low (figure 5c). By contrast, a larger $k_2$ increases both the fundamental and realized niches of type 1 and decreases both the fundamental and realized niches of type 2. Type 1 can expand the realized niche of type 2 by supplying $A_e$, especially when the growth of type 2 is significantly limited by abiotic reaction 2, which quickly consumes $A_e$.

Both compounds $A_e$ and C are the reactants of reaction 2. Larger $\Delta_f G_{A_e}^o$ and $\Delta_f G_C^o$ in $-\Delta_r G_2^o$ (see equations (2.4e) and (2.4f)) increase both the fundamental and realized niches of type 2, as shown in the lower panels of figure 5d,e, because a reaction converting an unstable compound (a compound with a relatively high $\Delta_f G^o$) to a more stable compound (a compound with a lower $\Delta_f G^o$) can generate more energy. A larger $\Delta_f G_C^o$ increases both the fundamental and realized niches of type 1 and the realized niche of type 2 (figure 5d). However, a larger $\Delta_f G_{A_e}^o$ decreases both the fundamental and realized niches of type 1 (figure 5e) because $A_e$ is the by-product of reaction 1 so that $-\Delta_r G_1^o$ decreases with increasing $\Delta_f G_{A_e}^o$ (see equation (2.4e)). The enhanced growth of type 2 from the use of $A_e$ with high $\Delta_f G_{A_e}^o$ may decrease the concentration of $A_e$, which can positively affect the growth of type 1 through the ARP. However, the overall $-\Delta_r G_1$ is insensitive to ARP when $-\Delta_r G_1^o$ is significantly low because of the low value of $\Delta_f G_{A_e}^o$. The different responses of types 1 and 2 to $\Delta_f G_{A_e}^o$ and $\Delta_f G_C^o$ suggest two conclusions.

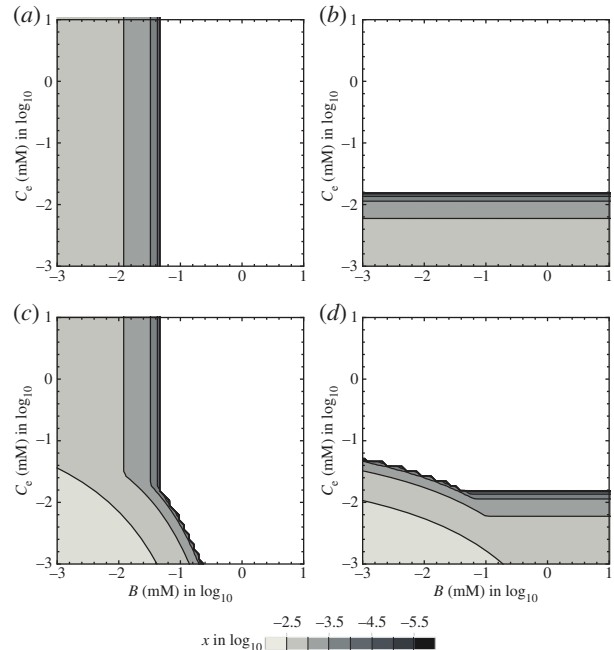

**Figure 4.** Steady-state responses on the $(B, C_e)$ plane. (a) Biomass of type 1 when it exists by itself. (b) Biomass of type 2 when it exists by itself. (c) Biomass of type 1 when it coexists with a mutualistic catabolic partner (type 2). (d) Biomass of type 2 when it coexists with a mutualistic catabolic partner (type 1). Parameters are set to the default values presented in electronic supplementary material, table S1.

First, types 1 and 2 do not readily recycle an element that is composed of compounds A and $A_e$ with a large gap between $\Delta_f G_A^o$ and $\Delta_f G_{A_e}^o$. Second, the presence of type 2 improves the fitness of type 1 more robustly when the standard Gibbs energy change of formation of a compound C is sufficiently higher than that of $C_e$, even when the concentration of B or $C_e$ may change. Consequently, microbes can recycle an element in two compounds with a large $\Delta_f G^o$ gap between the two. We note here that compounds B and C are not recycled between the two microbial types. A link between the microbial interaction and the Gibbs energy of a system will be discussed in §4d.

In a similar manner, we numerically calculated the niche expansion on a $(B_e, C)$ plane (see electronic supplementary material, appendix S4). We observed that the ARP encouraged the realized niche to expand in comparison with the model ignoring ARP, the latter being given by equation (2.3) with the second term on the right-hand side of equations (2.4c) and (2.4d) eliminated (electronic supplementary material, figure S1). Overall, the difference between the realized niche and the fundamental niche on the $(B_e, C)$ plane was greater in a model with the ARP term included than in the corresponding model with the ARP term ignored.

# 4. Discussion

## (a) One-way interaction and recycling interaction

In this paper, we proved that the mutualistic catabolic interaction between types 1 and 2 can increase the steady-state biomass of both types and the flow rate of an element that is included in the compounds catabolically recycling between them. In addition, type 1 (or 2) may expand the realized niche of type 2 (or 1).

Some of the results in this paper are similar to those in our previous model in which type 2 uses the catabolic by-product

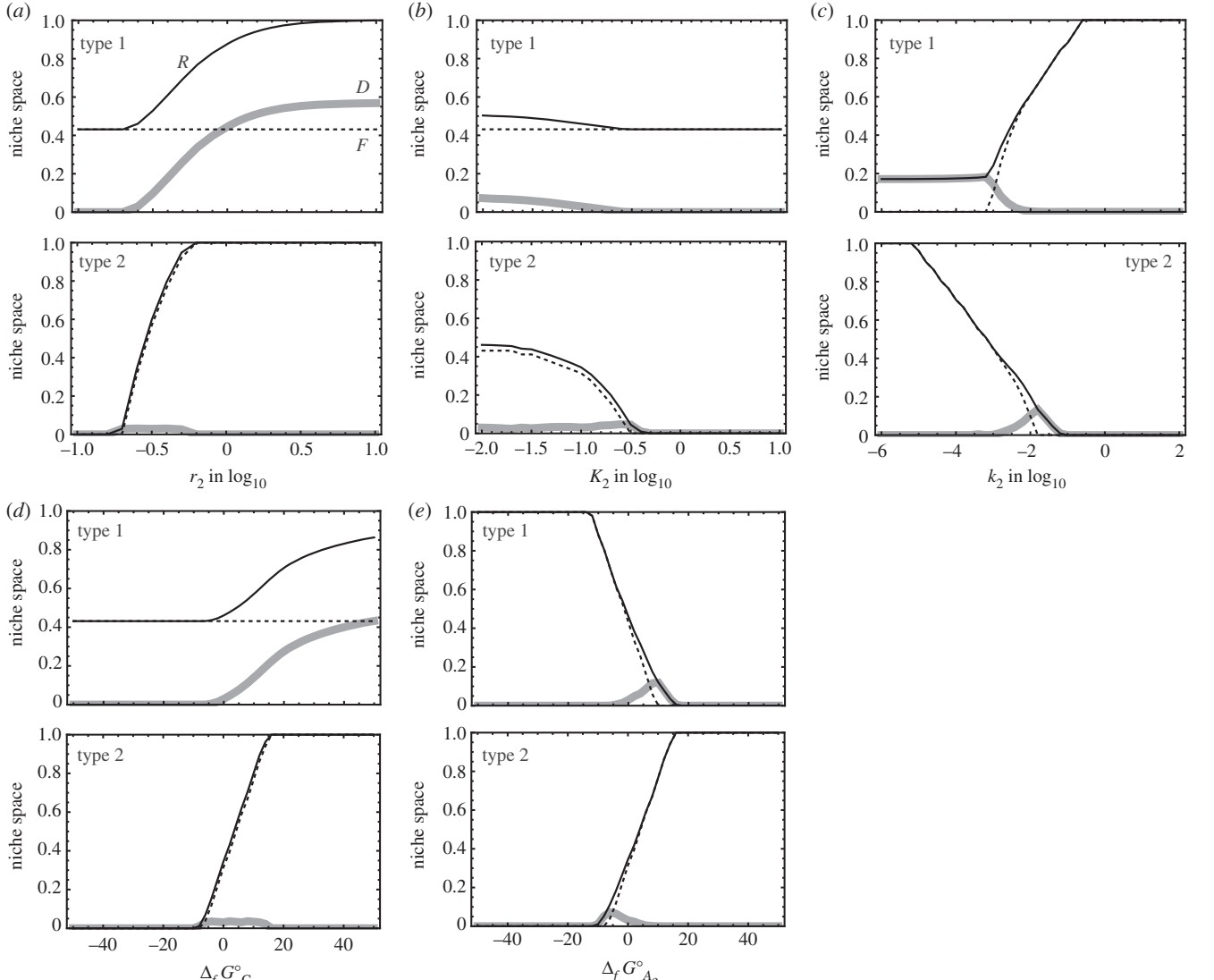

**Figure 5.** Parameter dependence of the size of the ARP-driven niche space for type 1 (upper panels) and type 2 (lower panels). The horizontal axes are on (a) $r_2$, (b) $K_2$, (c) $k_2$, (d) $\Delta_f G_C^0$ and (e) $\Delta_f G_{A_e}^0$ in the $(B, C_e)$ space. The solid black, dotted black and grey curves indicate the sizes of the realized niche, the fundamental niche and the difference between the two, respectively. The corresponding graphs for realized niche size, fundamental niche size and their difference are labelled by $R$, $F$ and $D$, respectively. The size of the niche space is defined as the number of sampled points belonging to the fundamental or realized niches divided by $51 \times 51$, where points are uniformly sampled within a square: $-3 < \ln B < 1$ and $-3 < \ln C_e < 1$ on the $(\ln B, \ln C_e)$ plane. See the text for further explanations. Other parameters are set to the default values presented in electronic supplementary material, table S1.

of type 1 but type 1 does not use the by-product of type 2 [18] (figure 6a). To clarify, we call the form of interaction between two catabolic reactions 'one-way interaction', as illustrated in figure 6a, where one reaction is upstream of the other. By contrast, we call the form of interaction illustrated in figure 6b 'recycling interaction', in which each reaction produces the by-product that is a reactant of the other. These two interactions are frequently observed in biogeochemical cycles.

From our previous report [18] and this paper, we can conclude that type 2 in the one-way interaction can increase the steady-state biomass of type 1 and expand the realized niche of type 1 only when the ARP is considered and sufficiently large, and type 2 in the recycle interaction can increase the steady-state biomass of type 1 and expand its realized niche regardless of the presence of the ARP. When the ARP is important, type 2 can further expand the realized niche of type 1. Table 1 summarizes and compares the effect of the presence of a catabolic partner with the other counterpart of one-way interaction and recycling interaction in figure 6.

## (b) Effect of model assumptions on the outcomes

We consider the situation in which the concentrations of B, $B_e$, C and $C_e$ are unchanged because of the presence of buffering effects. We performed a preliminary examination for the case where B, $B_e$, C and $C_e$ also change with the progress of reactions and observed; interestingly, the oscillatory fluctuation instead of predictable and steady-state behaviour reported in this paper, which will be discussed in future publication.

Unlike the Monod-type model, our model explicitly considers the density-dependent mortality because cells of the same catabolic type are often densely colonized in a consortium rather than they freely reside in a medium. Especially when type $i$ has high $r_i$ (the maximum catalytic rate per unit of biomass) and uses a reaction with high $-\Delta_r G_i^\circ$, the behaviour of the model became fragile, showing explosion, instead of the convergence to the model predictable and stable steady states. This implies that the spatial growth limitation on two microbes in a consortium would be important for maintaining the mutualistic interaction. To figure this

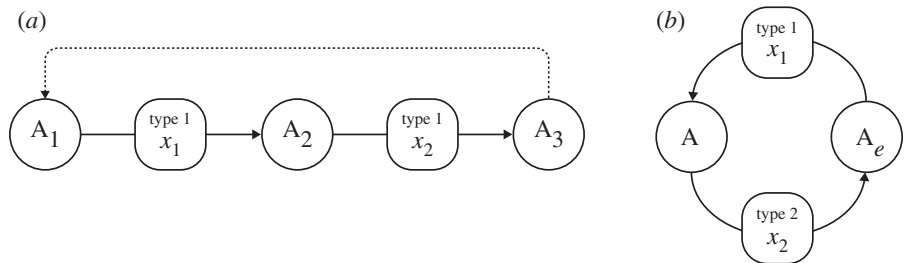

**Figure 6.** Type and example of mutualistic catabolic interactions. (a) One-way interaction studied in Seto & Iwasa [18]. (b) Recycling interaction studied in this paper.

**Table 1.** Effect of the presence of a catabolic partner to the other counterpart in figure 6. ++ indicates that the positive effect on the other counterpart increases in comparison with the case where the effect of ARP is ignored (the second term of the right-hand side of $-\Delta_r G$ is assumed to be 0).

| | one-way interaction | | recycling interaction | |
|---|---|---|---|---|
| | **without ARP** | **with ARP** | **without ARP** | **with ARP** |
| growth rate | (+, 0) | (++, +) | (+, +) | (++, ++) |
| steady-state biomass | (+, 0) | (++, +) | (+, +) | (++, ++) |
| reaction rate | (+, 0) | (++, +) | (+, +) | (++, ++) |
| realized niche | (+, 0) | (++, +) | (+, +) | (++, ++) |

out, the model needs to incorporate a spatial configuration of consortium and reaction–diffusion exports of cells and catabolic by-products in it.

When the model without the density-dependent mortality converged to a stable steady state, we confirmed that the results are similar to those summarized in table 1.

### (c) Material recycling and ecosystem productivity

The productivity of the terrestrial ecosystem is generally measured based on the ability of the ecosystem to sequester carbon, which is supported by the net primary production (net growth) of terrestrial plants. In the subsurface realm, both the availability of light and photosynthetic by-products (organic carbon) are limited, and primary productivity (carbon fixation) is supported by chemolithoautotrophs that harvest energy from redox reactions using only inorganic compounds and fix organic carbon from inorganic carbon ($CO_2$ or $HCO_3^-$). Accordingly, the productivity of the subsurface ecosystem depends on the net growth of the chemolithoautotrophic microbes. On the basis of our analysis, we conclude that mutualistic catabolic interactions increase the growth of microbes and thereby enhance the productivity of subsurface ecosystems.

The potentially positive effect of material cycling on ecosystem productivity has been discussed for the plant–decomposer interaction model, in which plants feed decomposers with litter and absorb inorganic nutrients (most likely nitrogen compounds) released by decomposers [21]. One important difference between the plant–decomposer interaction model and our model concerns the target processes in metabolism. Nitrogen is used for ATP-synthesizing processes (catabolism) and ATP-requiring processes (anabolism) and can limit the growth rate of microbes through both processes. In our model, as nitrogen compounds can be the electron–donor and/or acceptor compounds and are not accumulated as biomass, the availability of nitrogen compounds only limits ATP-synthesizing processes. To consider the energetic constraints on growth, we should take into account the $-\Delta_r G$ and ARP of the energy source reaction using a nitrogen compound. By contrast, in the plant–decomposer models, the availability of nitrogen is likely to limit the growth of plants and decomposers mainly through ATP-requiring processes because nitrogen is required not as the energy source but as the building block of biomass. However, we also note that catabolic and anabolic processes are inextricably linked. For example, the primary productivity of plants is often proportional to the nitrogen content of plant leaves, thus contributing to the rate of anabolism.

By considering the similarity of two models, we conclude that not the ARP, the characteristic term of our model, but the structure of recycling seems to have a key role in ecosystem productivity. Nevertheless, the ARP plays significant roles in niche expansion, as shown in §3c.

The other distinct characteristics of one-way interaction and recycling interaction models result in resource competition between abiotic processes and microbes. There are various abiotic processes that significantly affect the resource availability that controls the growth and fitness of organisms. For instance, under neutral pH conditions, the abiotic iron oxidation rate is so rapid that iron-oxidizing bacteria seem to have developed strategies to overcome the abiotic iron oxidation processes [8,22–26]. Our results suggest that making a consortium with coupling of different catabolic processes would potentially be beneficial to overcome the rapid abiotic reaction rates in ecosystems (lower panel in figure 5c).

### (d) Microbial community structure and Gibbs energy

Our analyses suggest that recycling interactions and potential niche expansion respond differently to the standard Gibbs

energy of the formation of compounds in reactions 1 and 2. When a type 1 microbe uses a reactant with a high $\Delta_f G^\circ$ that is not recycled between microbes ($B_e$ in this model), the realized niche of type 2 can be expanded because it can indirectly depend on reaction 3, the overall reaction of reactions 1 and 2, owing to the high productivity of reaction 1 that fuels reaction 2. However, when a type 1 microbe uses a reactant with a high $\Delta_f G^\circ$ that is recycled between microbes (A in this model), type 1 cannot build a mutually beneficial interaction with type 2 because the lower energy production from reaction 2 may not support the growth of type 2. This suggests two considerations. First, the consortium of types 1 and 2 may be able to increase their realized niches when $-\Delta_r G^\circ$ of reaction 3 (see equation (4.1c)), the overall reaction of reactions 1 and 2 is high. Second, the availability of compounds and the possible redox reactions or the availability of Gibbs energy in a system may determine microbial interaction types because recycling catabolic interaction is unlikely to occur without the presence of an energetically suitable electron donor or acceptor (potentially C or $B_e$ in our model) when an element in two compounds has a large $\Delta_f G^\circ$ gap. One example is the difficulty of the complete catabolic recycle between carbon dioxide with $\Delta_f G^\circ = -394.36$ kJ mol$^{-1}$ and glucose with $\Delta_f G^\circ = -915.29$ kJ mol$^{-1}$, or $-152.55$ kJ per 1 mole of carbon. The recycle between carbon dioxide and glucose can be achieved by the presence of phototrophic organisms that capture sunlight to force electrons away from water molecule which eventually power the synthesis of glucose (figure 1a), whereas the complete catabolic recycle between carbon dioxide and methane with $\Delta_f G^\circ = -375.56$ kJ mol$^{-1}$ is frequently observed in subsurface ecosystems (figure 1b). The relationship between biodiversity, community structure and ecosystem functioning has been one of the central issues in ecological studies. Our study may provide a key insight into how the Gibbs energy available in a system can determine the interaction types among microbes and ecosystem expansion in subsurface ecosystems.

## (e) ARP-driven expansion of the geochemical niche

Our results showed that the realized niche of type 1 (or 2) in the $(B, C_e)$ space and the $(B_e, C)$ space was expanded in the presence of a mutualistic catabolic partner. The overall reaction of reactions 1 and 2 is

$$B_e + C \rightarrow B + C_e \quad \text{(Reaction 3).} \tag{4.1a}$$

The negative of the Gibbs energy change caused by reaction 3 is

$$-\Delta_r G_3 = -\Delta_r G_3^\circ + RT \ln \frac{B_e C}{B C_e}, \tag{4.1b}$$

where $-\Delta_r G_3^\circ$ is

$$-\Delta_r G_3^\circ = -(\Delta_f G_B^\circ + \Delta_f G_{C_e}^\circ) + (\Delta_f G_{B_e}^\circ + \Delta_f G_C^\circ). \tag{4.1c}$$

Accordingly, $-\Delta_r G_3$ is equivalent to the sum of $-\Delta_r G_1$ and $-\Delta_r G_2$.

$$-\Delta_r G_3 = -\Delta_r G_1 + (-\Delta_r G_2). \tag{4.1d}$$

We may be able to consider that the consortium of types 1 and 2 collectively harvests their energy from reaction 3. This could be a significant benefit for microbes harnessing a reaction with low $-\Delta_r G^\circ$. The growth of these microbes is often limited in the place where the ARP of its energy-harvesting reaction is maintained higher than that in other environments. The presence of mutualistic catabolic partners can enable these microbes to be freed from energetic constraints and to invade other systems by depending on reaction 3.

Especially for microbes harnessing reactions with low $-\Delta_r G^\circ$ the ARP may play indispensable roles in potential growth and survival and community structure. Because the low availability of oxygen and organic carbon on the early Earth limited the favourable (feasible) redox reactions as being energy sources, the ARP should have affected the development of microbial ecosystems on the early Earth where the $-\Delta_r G^\circ$ of possible redox reactions were generally orders of magnitude lower than those under current Earth conditions.

Data accessibility. Example source file is provided as the electronic supplementary material.

Authors' contributions. M.S. and Y.I. designed research, performed research and wrote the paper.

Competing interests. We declare we have no competing interests.

Funding. This work was supported by JSPS Grant-in-Aid for Scientific Research (C) (grant no. 19K06853] to M.S.

Acknowledgements. We thank the following people for their very useful comments: K. Kadowaki, K. Koba, M. Kondoh, Y. Tachiki and K. Uriu. We are also thankful to two anonymous reviewers for their constructive comments.

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
