## [Reviewer comments · Proceedings of the Royal Society B: Biological Sciences]

Review History

RSPB-2020-0610.R0 (Original submission)

Review form: Reviewer 1

Recommendation

Accept with minor revision (please list in comments)

Scientific importance: Is the manuscript an original and important contribution to its field?

Good

General interest: Is the paper of sufficient general interest?

Good

Quality of the paper: Is the overall quality of the paper suitable?

Good

Is the length of the paper justified?

No

Should the paper be seen by a specialist statistical reviewer?

No

Do you have any concerns about statistical analyses in this paper? If so, please specify them explicitly in your report.

No

It is a condition of publication that authors make their supporting data, code and materials available - either as supplementary material or hosted in an external repository. Please rate, if applicable, the supporting data on the following criteria.

Is it accessible?

Yes

Is it clear?

No

Is it adequate?

No

Do you have any ethical concerns with this paper?

No

Comments to the Author

Review of "Microbial material cycling, energetic constraints and ecosystem expansion in subsurface ecosystems" by Mayumi Seto and Yoh Iwasa

The authors studied a simple, tractable system of mutualistic metabolic types involved in a cycle, which is isolated from the 'rest of the ecosystem' assuming steady state for other reactants. The model and results are clearly described and sound. I like the inclusion of the chemical reactions competing with the microbially catalyzed reactions. This study builds on their previous work (ref 18) of 'one way interaction' by now looking at cyclic interactions. This is the novelty of this submission.

The model assumes a density dependent mortality of the microbes, this is rather unusual for such models so it would be good to motivate this choice better and study its effect. This density self-regulation is a different approach, common in ecology, to the more mechanistic rest of the model, common in microbial ecology.

Because B and C are not recycled in the model, it is not clear to me whether the conclusions about the effect of high Gibbs energies of formation for B or C are just the result of the assumption that these compounds are external and in steady state?

Similarly, regarding the conclusion that recycling reactions are unlikely to occur when an element in a compound with significantly low or high Gibbs energy of formation is recycled (Line 407) made me wonder if there is an example of this? And if not, why not? Are there no such elements, or is it that cycles are more complex and interconnected with others?

A table and/or schematic to summarize the results would help, it is easy to get lost in the details and not remember a thing at the end. If possible, the table could include results of the previous work for comparison.

Minor comments

1. Abstract is hard to understand without having read their previous work, eg try to briefly explain the 'abundant resource premium' and the effect of Gibbs energy. I notice that this ARP is explained in the introduction, and apparently a term they coined in their previous studies, but it is hard to guess what it means from the words used so in the abstract it leads to frowns.
2. The introduction into microbial energy metabolism could be more accurate, many statements are true for certain cases but not in general and this isn't made clear. Make clear whether a

statement is generally true, or for a particular (set of) example. For example, some of the statements made for aerobic respiration are not limited to aerobic respiration. Anaerobic respiration and fermentation are not properly considered and the carbon cycle was running for billions of years before oxygen. Iron can be oxidized without oxygen as acceptor (phototrophically or reducing nitrate).

3. Line 73: 'when one mole' is misleading in the sense that two mole would provide more energy, rewrite
4. Line 111: replace 'two different compounds composed' with 'two different redox states'
5. Line 159: energy gain here probably means ATP gain? and is different from Gibbs energy change, please make clear in the manuscript
6. Line 160: 'useful energy' should be more appropriately worded, eg energy that can perform chemical work
7. Line 184: the single locally stable steady state is there not a trivial steady state too?
8. Line 316: make clearer what a significantly high or low Gibbs energy of formation would be (examples where this is the case or numbers)
9. Line 334 introduced the geochemical niche without making clear how this is related to fundamental/realized niche
10. Line 354 some organic carbon is typically available in the subsurface, 'imported' from above
11. Line 379: which two models are being compared here?
12. Line 434 and following: I don't understand how the energetics of reactions has changed due to changes on Earth, do the authors mean that with the availability of oxygen, other redox processes are possible (while the same reaction does not change energetics)?
13. Fig. 1. I don't understand the meaning/purpose of the arrows directly from oxygen or sulfate to Fe(III) or CO₂
14. Fig. 3 legend, the difference between panels a, b and c should be better explained, not just list all parameters, say what is different. And explain better what is shown, label the curves in the figure so you don't have to say in the text 'the solid line with positive slope is ...' A lot of the main text can be cut if the figure is labelled and explained in the panels and legend. As it is, one has to flip back and forth between text and figure.

Review form: Reviewer 2

Recommendation

Accept with minor revision (please list in comments)

Scientific importance: Is the manuscript an original and important contribution to its field?

Good

General interest: Is the paper of sufficient general interest?

Acceptable

Quality of the paper: Is the overall quality of the paper suitable?

Good

Is the length of the paper justified?

Yes

Should the paper be seen by a specialist statistical reviewer?

No

Do you have any concerns about statistical analyses in this paper? If so, please specify them explicitly in your report.

No

It is a condition of publication that authors make their supporting data, code and materials available - either as supplementary material or hosted in an external repository. Please rate, if applicable, the supporting data on the following criteria.

Is it accessible?

Yes

Is it clear?

Yes

Is it adequate?

Yes

Do you have any ethical concerns with this paper?

No

Comments to the Author

This manuscript develops a model on a catabolic cycle between two types of microbes and analyses its effect on their population dynamics from the perspective of thermodynamics. This study shows that their biomass, efficiency of resource utilization, and ecological niches are improved by the symbiotic interaction. Based on these results, this study argues that the first two results are caused by the cyclic structure of the catabolic interaction, while the last one is derived from stoichiometric effects of the interaction. Also, this study provides insights into what structure of catabolic interaction can be possible in ecological community from thermodynamical perspective.

Although their results that the symbiotic interaction can improve the biomass and ecological niches of microbes are not so surprising, I thought that their above discussions are very insightful, and they can provide a new perspective on studies of ecological community. Such discussion indicates that their rigorous modelling beyond intuition is worthwhile. I also thought that the model is appropriate and mostly described.

My following comments are all fairly minor:

Introduction: One of the famous hypotheses of the origin of eukaryotes and mitochondria is based on a catabolic cycle (syntrophic hypothesis: Moreira & López-García (1998) *J. Mol. Evol.* 47, 517–530). This might highlight the importance of this study from not only ecological but also evolutionary aspects.

L 78: The gas constant, R , should have the dimension of T^{-1} .

L 124 & 128: For simplification, it is assumed that A is completely recycled and concentrations of all other compounds are constant. I felt such assumptions are suitable as a first step of modelling, while it would be good if the authors touched their evaluation how results are affected by relaxing the assumptions.

Eq 4a: I think there are some missing parentheses here?

L 140: If it is possible, it would be nice if there is an intuitive explanation of the Monod equation or the multiplicative form of Michaelis–Menten equation.

L 272: I felt the explanation that there is no conflict between the two species as they are mutualist is not appropriate. The reason is that they do not compete for the same resources, isn't it?

Fig. 4: Although I expected the case where both species cannot invade and exist without each other, such mutual dependency seems to not occur. This model does not have multistability (L 184). Is it hard to occur in principle, or is it possible by relaxing some of the assumptions? Since such a situation highlights the importance of catabolic interactions, if the authors have any idea about it, it would be nice if the difficulty or possibility of such mutual dependency is touched.

L 391: Fig. 5© → Fig. 5C

L 398: reaction 3 → reaction 2?

L 409: This argument is interesting.

Decision letter (RSPB-2020-0610.R0)

19-Jun-2020

Dear Dr Seto:

Your manuscript has now been peer reviewed and the reviews have been assessed by an Associate Editor. The reviewers' comments (not including confidential comments to the Editor) and the comments from the Associate Editor are included at the end of this email for your reference. As you will see, the reviewers and the Editors have raised some concerns with your manuscript and we would like to invite you to revise your manuscript to address them.

Research ethics:

Use of animals and field studies:

It is a condition of publication that you make available the data and research materials supporting the results in the article. Datasets should be deposited in an appropriate publicly available repository and details of the associated accession number, link or DOI to the datasets must be included in the Data Accessibility section of the article

(<https://royalsociety.org/journals/ethics-policies/data-sharing-mining/>). Reference(s) to datasets should also be included in the reference list of the article with DOIs (where available).

Please submit a copy of your revised paper within three weeks. If we do not hear from you within this time your manuscript will be rejected. If you are unable to meet this deadline please let us know as soon as possible, as we may be able to grant a short extension.

Best wishes,

Professor Gary Carvalho

Associate Editor

Board Member: 1

Comments to Author:

Thank you for submitting your manuscript to Proceedings B. I enjoyed reading this article, and the two expert reviewers have also provided a positive assessment of your manuscript. The reviewers have highlighted some relatively minor revisions that would be required before the manuscript can be accepted, and I would encourage you to address these fully in the revised version. Reviewer 1 found that the results are very detailed in places, and can be difficult to follow, and suggested the addition of a summary table and schematic visual representation of the trends in the results (e.g. niche expansion) to assist with clarity. Both reviewers have also provided a list of specific comments to address (provided below), and reviewer 1 raises some specific elements of the model and its interpretation that must be addressed and justified in the revision. Thanks once again for your submission, and I look forward to receiving the revised version.

Reviewer(s)' Comments to Author:

Referee: 1

Comments to the Author(s)

Review of "Microbial material cycling, energetic constraints and ecosystem expansion in subsurface ecosystems" by Mayumi Seto and Yoh Iwasa

The authors studied a simple, tractable system of mutualistic metabolic types involved in a cycle, which is isolated from the 'rest of the ecosystem' assuming steady state for other reactants. The model and results are clearly described and sound. I like the inclusion of the chemical reactions competing with the microbially catalyzed reactions. This study builds on their previous work (ref 18) of 'one way interaction' by now looking at cyclic interactions. This is the novelty of this submission.

The model assumes a density dependent mortality of the microbes, this is rather unusual for such models so it would be good to motivate this choice better and study its effect. This density self-regulation is a different approach, common in ecology, to the more mechanistic rest of the model, common in microbial ecology.

Because B and C are not recycled in the model, it is not clear to me whether the conclusions about the effect of high Gibbs energies of formation for B or C are just the result of the assumption that these compounds are external and in steady state?

Similarly, regarding the conclusion that recycling reactions are unlikely to occur when an element in a compound with significantly low or high Gibbs energy of formation is recycled (Line 407) made me wonder if there is an example of this? And if not, why not? Are there no such elements, or is it that cycles are more complex and interconnected with others?

A table and/or schematic to summarize the results would help, it is easy to get lost in the details and not remember a thing at the end. If possible, the table could include results of the previous work for comparison.

Minor comments

1. Abstract is hard to understand without having read their previous work, eg try to briefly explain the 'abundant resource premium' and the effect of Gibbs energy. I notice that this ARP is explained in the introduction, and apparently a term they coined in their previous studies, but it is hard to guess what it means from the words used so in the abstract it leads to frowns.
2. The introduction into microbial energy metabolism could be more accurate, many statements are true for certain cases but not in general and this isn't made clear. Make clear whether a

statement is generally true, or for a particular (set of) example. For example, some of the statements made for aerobic respiration are not limited to aerobic respiration. Anaerobic respiration and fermentation are not properly considered and the carbon cycle was running for billions of years before oxygen. Iron can be oxidized without oxygen as acceptor (phototrophically or reducing nitrate).

3. Line 73: 'when one mole' is misleading in the sense that two mole would provide more energy, rewrite
4. Line 111: replace 'two different compounds composed' with 'two different redox states'
5. Line 159: energy gain here probably means ATP gain? and is different from Gibbs energy change, please make clear in the manuscript
6. Line 160: 'useful energy' should be more appropriately worded, eg energy that can perform chemical work
7. Line 184: the single locally stable steady state is there not a trivial steady state too?
8. Line 316: make clearer what a significantly high or low Gibbs energy of formation would be (examples where this is the case or numbers)
9. Line 334 introduced the geochemical niche without making clear how this is related to fundamental/realized niche
10. Line 354 some organic carbon is typically available in the subsurface, 'imported' from above
11. Line 379: which two models are being compared here?
12. Line 434 and following: I don't understand how the energetics of reactions has changed due to changes on Earth, do the authors mean that with the availability of oxygen, other redox processes are possible (while the same reaction does not change energetics)?
13. Fig. 1. I don't understand the meaning/purpose of the arrows directly from oxygen or sulfate to Fe(III) or CO₂
14. Fig. 3 legend, the difference between panels a, b and c should be better explained, not just list all parameters, say what is different. And explain better what is shown, label the curves in the figure so you don't have to say in the text 'the solid line with positive slope is ...' A lot of the main text can be cut if the figure is labelled and explained in the panels and legend. As it is, one has to flip back and forth between text and figure.

Referee: 2

Comments to the Author(s)

This manuscript develops a model on a catabolic cycle between two types of microbes and analyses its effect on their population dynamics from the perspective of thermodynamics. This study shows that their biomass, efficiency of resource utilization, and ecological niches are improved by the symbiotic interaction. Based on these results, this study argues that the first two results are caused by the cyclic structure of the catabolic interaction, while the last one is derived from stoichiometric effects of the interaction. Also, this study provides insights into what structure of catabolic interaction can be possible in ecological community from thermodynamical perspective.

Although their results that the symbiotic interaction can improve the biomass and ecological niches of microbes are not so surprising, I thought that their above discussions are very insightful, and they can provide a new perspective on studies of ecological community. Such discussion indicates that their rigorous modelling beyond intuition is worthwhile. I also thought that the model is appropriate and mostly described.

My following comments are all fairly minor:

Introduction: One of the famous hypotheses of the origin of eukaryotes and mitochondria is based on a catabolic cycle (syntrophic hypothesis: Moreira & López-García (1998) *J. Mol. Evol.* 47, 517–530). This might highlight the importance of this study from not only ecological but also evolutionary aspects.

L 78: The gas constant, R, should have the dimension of T^{-1} .

L 124 & 128: For simplification, it is assumed that A is completely recycled and concentrations of all other compounds are constant. I felt such assumptions are suitable as a first step of modelling, while it would be good if the authors touched their evaluation how results are affected by relaxing the assumptions.

Eq 4a: I think there are some missing parentheses here?

L 140: If it is possible, it would be nice if there is an intuitive explanation of the Monod equation or the multiplicative form of Michaelis–Menten equation.

L 272: I felt the explanation that there is no conflict between the two species as they are mutualist is not appropriate. The reason is that they do not compete for the same resources, isn't it?

Fig. 4: Although I expected the case where both species cannot invade and exist without each other, such mutual dependency seems to not occur. This model does not have multistability (L 184). Is it hard to occur in principle, or is it possible by relaxing some of the assumptions? Since such a situation highlights the importance of catabolic interactions, if the authors have any idea about it, it would be nice if the difficulty or possibility of such mutual dependency is touched.

L 391: Fig. 5© → Fig. 5C

L 398: reaction 3 → reaction 2?

L 409: This argument is interesting.

Author's Response to Decision Letter for (RSPB-2020-0610.R0)

See Appendix A.

Decision letter (RSPB-2020-0610.R1)

07-Jul-2020

Dear Dr Seto

I am pleased to inform you that your manuscript entitled "Microbial material cycling, energetic constraints and ecosystem expansion in subsurface ecosystems" has been accepted for publication in Proceedings B.

Open Access

Paper charges

Sincerely,

Professor Gary Carvalho

Associate Editor:

Comments to Author:

Thank you for providing a comprehensive revision of the manuscript that addresses all of the reviewers and AE comments provided in the previous round of review. Congratulations, and thank you for submitting your work to Proceedings B.

Appendix A

List of changes we made and our replies to the comments by the associate editor and reviewers:

(Numbers indicate the sections in their letters.)

We are grateful to the editor and two reviewers for their detailed and very useful comments. We revised the paper in responding to each of the comments by the editor and reviewers.

Reply to comments from associate editor:

> Thank you for submitting your manuscript to *Proceedings B*. I enjoyed reading this article, and
> the two expert reviewers have also provided a positive assessment of your manuscript. The
> reviewers have highlighted some relatively minor revisions that would be required before
> the manuscript can be accepted, and I would encourage you to address these fully in the
> revised version. Reviewer 1 found that the results are very detailed in places, and can be
> difficult to follow, and suggested the addition of a summary table and schematic visual
> representation of the trends in the results (e.g. niche expansion) to assist with clarity. Both
> reviewers have also provided a list of specific comments to address (provided below),
> and reviewer 1 raises some specific elements of the model and its interpretation that
> must be addressed and justified in the revision. Thanks once again for your submission, and I
> look forward to receiving the revised version.

[Answer] Thank you very much for handling our manuscript. We fully revised our manuscript as advised by the reviewers 1 and 2. Please see our reply to the comments from the Reviewer 1 and 2, below.

Reply to comments from reviewer 1:

> The authors studied a simple, tractable system of mutualistic metabolic types involved in a
> cycle, which is isolated from the 'rest of the ecosystem' assuming steady state for other
> reactants. The model and results are clearly described and sound. I like the inclusion of
> the chemical reactions competing with the microbially catalyzed reactions. This study
> builds on their previous work (ref 18) of 'one way interaction' by now looking at cyclic
> interactions. This is the novelty of this submission.

[Answer] We are very thankful to reviewer #1 for detailed comments and many useful advices.

[R1-1] > The model assumes a density dependent mortality of the microbes, this is rather unusual
> for such models so it would be good to motivate this choice better and study its effect. This
> density self-regulation is a different approach, common in ecology, to the more mechanistic
> rest of the model, common in microbial ecology.

[Answer] The density dependence represents the setting of the model in which microbes are not distributed uniformly over the whole system but rather they form a tight consortium in which cells are in contact with neighbors. Within this consortium, two microbes are separated forming clusters of the same type, although the clusters of x_1 and those of x_2 are not very far from each other, allowing them to interact with each other by exchanging catabolic by-products. The model in Eqs. (4b) and (4c) represent this spatial configuration in a simplified manner. The microbes engage in the strong competitive interaction within the same species (but not between species). As they are spatially very close, consuming resources would reduce the availability for nearby conspecifics. In

addition, they also engage in interaction by exchanging catabolic by-products by the presence of these microbes at an intermediate distance.

If we removed the intraspecific competition terms in Eqs. (4b) and (4c), the behavior of the model became very fragile, showing explosion, instead of the convergence to the predictable and stable stationary states. This suggests us that the spatial configuration of forming consortium of two types of bacteria is important for proper functioning of the system.

In the revised version we have:

- [1] Before presenting the equations for the model, we described the situation represented by the model (In lines 113 – 117),
- [2] In lines 157 – 160, we pointed out that the terms of intraspecific inhibition represent the spatial configuration of forming tight consortium, instead of interaction in perfectly mixed medium.
- [3] In the discussion section, we pointed out this and stated that the model without this inhibition does not have the steady state property we discussed in this paper, which suggests that importance of this spatial arrangement of the microbes for maintaining this interaction. We also pointed out that a more realistic and extensive mathematical studies might include spatial configuration of consortium handled by reaction-diffusion model, which would be much more complex than the current paper. (lines 373 – 384)

[R1-2] > *Because B and C are not recycled in the model, it is not clear to me whether the > conclusions about the effect of high Gibbs energies of formation for B or C are just the > result of the assumption that these compounds are external and in steady state?*

[Answer] We consider the situation in which the concentrations of B and C are strongly buffered in a much greater medium, and hence we can assume that these values are constant in the model. In lines 140 – 141, we added a new sentence and clearly stated this.

We also performed a preliminary examination of the case including the dynamics of B and C, and the model's behavior was not predictable, including oscillatory fluctuation, instead of predictable and steady state behavior reported in this paper.

In the discussion section 4.2, in lines 367 – 372, we pointed out this and mentioned that studying the case with neither intraspecific self-regulation nor buffering B and C might be an interesting topic of future theoretical study.

[R1-3] > *Similarly, regarding the conclusion that recycling reactions are unlikely to occur when > an element in a compound with significantly low or high Gibbs energy of formation is > recycled (Line 407) made me wonder if there is an example of this? And if not, why not? > Are there no such elements, or is it that cycles are more complex and interconnected with others?*

[Answer] We appreciate this important comment. We found that the discussion for this result was ambiguous and misleading. In lines 326 – 334, we rephrased the interpretation of the results as follows:

“First, types 1 and 2 do not readily recycle an element that is composed of compounds A and A_e with a large gap between $\Delta_f G_A^o$ and $\Delta_f G_{A_e}^o$. Second, the presence of type 2 improves the fitness of type 1 more robustly when the standard Gibbs energy change of formation of a compound C is sufficiently higher than that of C_e, even when the

concentration of B or C_e may change. Consequently, microbes can recycle an element in two compounds with a large $\Delta_f G^\circ$ gap between the two. We note here that compounds B and C are not recycled between the two microbial types. A link between the microbial interaction and the Gibbs energy of a system will be discussed later in Section 4.4.”

We here emphasized that not the values of $\Delta_f G^\circ$ of A and A_e but the gap of $\Delta_f G^\circ$ between A and A_e affects the achievement of recycle between A and A_e .

In lines 447 – 453, we also added an example for this:

“One example is the difficulty of the complete catabolic recycle between carbon dioxide with $\Delta_f G^\circ = -394.36 \text{ kJ mol}^{-1}$ and glucose with $\Delta_f G^\circ = -915.29 \text{ kJ mol}^{-1}$. The recycle between carbon dioxide and glucose can be achieved by the presence of phototrophic organisms that capture sunlight to force electrons away from water molecule which eventually power the synthesis of glucose (Fig. 1(a)), whereas the complete catabolic recycle between carbon dioxide and methane with $\Delta_f G^\circ = -375.56 \text{ kJ mol}^{-1}$ is frequently observed in subsurface ecosystems (Fig. 1(b)).”

[R1-4] > *A table and/or schematic to summarize the results would help, it is easy to get lost in > the details and not remember a thing at the end. If possible, the table could include results of > the previous work for comparison.*

[Answer] Following the advice of this reviewer, we made Table 1 to summarize the results.

Minor comments:

[R1-5] > *Abstract is hard to understand without having read their previous work, eg try to briefly > explain the ‘abundant resource premium’ and the effect of Gibbs energy. I notice that this ARP > is explained in the introduction, and apparently a term they coined in their previous studies, but > it is hard to guess what it means from the words used so in the abstract it leads to frowns.*

[Answer] Following this advice, in lines 27 – 32, the corresponding sentences can now read:

“By coupling two catabolic reactions, types 1 and 2 can also expand their realised niches through the abundant resource premium, the effect of relative quantities of products and reactants on the available chemical energy, which is especially important for microbes under strong energetic limitations. The plausibility of mutually beneficial interactions is controlled by the available chemical energy (Gibbs energy) of the system.”

[R1-6] > *The introduction into microbial energy metabolism could be more accurate, many statements > are true for certain cases but not in general and this isn’t made clear. Make clear whether a statement > is generally true, or for a particular (set of) example. For example, some of the statements made for > aerobic respiration are not limited to aerobic respiration. Anaerobic respiration and fermentation are > not properly considered and the carbon cycle was running for billions of years before oxygen. Iron can > be oxidized without oxygen as acceptor (phototrophically or reducing nitrate).*

[Answer] We are thankful to this important comment. To avoid the ambiguity, we tried to choose more accurate words to describe the microbial energy metabolism.

In lines 37 – 39,

“In terrestrial ecosystems, the carbon cycle between carbon dioxide and organic matter under aerobic condition is predominantly accomplished by plants and decomposers (or aerobic respiratory organisms)”

We added the underlined words so that one can recognize that we explain the case of aerobic condition. As this referee mentioned, some of the aspects of energy metabolism of aerobic respiration are not limited to aerobic respiration. However, we think those statements are not necessary for the introduction of this paper. We are now preparing a review paper which aims to introduce more detailed microbial energy metabolism to biologists and ecologists.

In lines 46 – 48,

“Electron transfer establishes an electrochemical gradient on the cell membrane, which eventually powers ATP synthesis (more specifically ATP synthesis by the oxidative phosphorylation) [1,2]”

We here noted that the electrochemical gradient powers ATP synthesis by the oxidative phosphorylation.

In lines 49 – 50, we replaced “We focus here on the material cycles between two alternative forms of an element, which are catalysed ...” with “We focus here on the material cycles between two alternative forms of an element, which can be achieved ...” so that we do not deny the cycling via the phototrophic or heterotrophic reactions.

In lines 53 – 57,

“The iron cycle driven by two possible catabolic reactions is illustrated in Fig. 1(b). Some bacteria harvest energy from iron oxidation by coupling Fe(II) and oxygen (or an alternative electron-acceptor compound), in which electrons are transferred from Fe(II) to oxygen and Fe(III) is produced.”

The amended sentences state that iron oxidation occurs by utilizing oxygen or an alternative electron-acceptor compound.

[R1-7] > Line 73: ‘when one mole’ is misleading in the sense that two mole would provide more > energy, rewrite

[Answer] In lines 76 – 78, the corresponding sentence was rewritten as follows: “the negative of the Gibbs energy change of a reaction ($-\Delta_r G$ in kJ mol^{-1}) provides the maximum available energy per unit of reaction progress (i.e., when one mole of a catabolic reaction occurs)”

[R1-8] > Line 111: replace ‘two different compounds composed’ with ‘two different redox states’

[Answer] We would like to use ‘compounds’ because most biologists and ecologists are not very familiar with ‘redox states’ (note that main targets of this article are ecologists). In lines 123 – 124, we rewrite the corresponding part as follows:

“two different compounds, or two different redox states,”

[R1-9] > *Line 159: energy gain here probably means ATP gain? and is different from Gibbs energy > change, please make clear in the manuscript*

[Answer] In lines 172 – 173, the corresponding part was rewritten as follows:
“ q_i is the biomass yield of species i for a given energy gain from ATP that is generated from a catabolic reaction”

[R1-10] > *Line 160: ‘useful energy’ should be more appropriately worded, eg energy that can > perform chemical work*

[Answer] Following the advice, in lines 173 – 174, the corresponding part was amended as follows:

“...the fraction of energy that can be utilized for ATP synthesis ...”

[R1-11] > *Line 184: the single locally stable steady state is there not a trivial steady state too?*

[Answer] We can prove that, for any choice of parameters, there exists one single "locally stable" steady state. This locally stable steady state can be a trivial steady state or a nontrivial one. A trivial steady state always exists, but it becomes "unstable" if another steady state (with positive microbial abundance) exists and is stable. Please see the Appendix S3 (Local stability of the candidate steady state) for mathematical arguments supporting this statement.

[R1-12] > *Line 316: make clearer what a significantly high or low Gibbs energy of formation > would be (examples where this is the case or numbers)*

[Answer] Please see the reply to R1-3.

[R1-13] > *Line 334 introduced the geochemical niche without making clear how this is related to > fundamental/realized niche*

[Answer] In line 348, to avoid ambiguity, "geochemical" was replaced with "realized".

[R1-14] > *Line 354 some organic carbon is typically available in the subsurface, ‘imported’ from > above*

[Answer] In lines 389 – 390, to be more accurate, the corresponding sentence was amended as follows:

“both the availability of light and photosynthetic by-products (organic carbon) are limited”

[R1-15] > *Line 379: which two models are being compared here?*

[Answer] In lines 420 – 421, two models are replaced by “one-way interaction and recycling interaction models”.

[R1-16] > *Line 434 and following: I don’t understand how the energetics of reactions has changed > due to changes on Earth, do the authors mean that with the availability of oxygen, other redox > processes are possible (while the same reaction does not change energetics)?*

[Answer] In lines 479 – 483, we amended the corresponding sentence as follows:

“Because the low availability of oxygen and organic carbon on the early Earth limited the favourable (feasible) redox reactions as being energy sources, the ARP should have affected the development of microbial ecosystems on the early Earth where $-\Delta_r G^\circ$ of possible redox reactions were generally orders of magnitude lower than those under current surface Earth conditions.”

[R1-17] > Fig. 1. I don't understand the meaning/purpose of the arrows directly from oxygen > or sulfate to Fe(III) or CO2

[Answer]: Following the advice of this reviewer, we amended the Figure 1 and its caption. We added the following sentence in the caption of Fig. 1:

“Fe(II) can be Fe²⁺, Fe(II) complex, or Fe(II) oxides/hydroxides and Fe(III) can be Fe³⁺, Fe(III) complex, or Fe(III) oxides/ hydroxides.”

[R1-18] > Fig. 3 legend, the difference between panels a, b and c should be better explained, not just > list all parameters, say what is different. And explain better what is shown, label the curves in > the figure so you don't have to say in the text 'the solid line with positive slope is ...' A lot of > the main text can be cut if the figure is labelled and explained in the panels and legend. As it > is, one has to flip back and forth between text and figure.

[Answer] We are thankful to this important comment. Following the advice, we wrote the details in the figure caption. The figure caption can now read:

“**Figure 3.** The reaction rate from A to A_e and that from A_e to A (upper panels) and the biomass contents of two microbes (lower panels) for the cases where the coexisting steady state ($\hat{A}, \hat{x}_1, \hat{x}_2$) is the only locally stable steady state while each Type 1 or 2 can solely invade the system for (a) and (b) and while only Type 2 can solely invade the system for (c). The only difference between (a) and (b) is the values of $-\Delta_r G_1^\circ$ and $-\Delta_r G_2^\circ$, which highlights the effect of relative significance of ARP (high for (a) and low for (b)) on the steady state. The horizontal axis shows the molar concentration of A. The two solid curves in the upper panels are $\psi_1(A) = \psi_1(A, \varphi_1(A))$ and $\psi_2(A) = \psi_2(A, \varphi_2(A))$, defined as Eqs. (7a) and (7b), indicating the rate of reaction from A to A_e and the rate of reaction from A_e to A, respectively. The two broken lines are $k_1 A B_e$ and $k_2 (A_T - A) C$, indicating the rates of two abiotic reactions. The two curves in the lower panels are $\varphi_1(A)$ and $\varphi_2(A)$, defined as Eqs. (6a) and (6b). There are for possible steady states: neither Type 1 nor 2 exists at $(A_0, 0, 0)$; Type 1 only exists at $(\bar{A}_1, \bar{x}_1, 0)$; Type 2 only exists at $(\bar{A}_2, 0, \bar{x}_2)$; Both Type 1 and 2 exist at $(\hat{A}, \hat{x}_1, \hat{x}_2)$. R_0, R_1, R_2 , and \hat{R} correspond to the steady state reaction rate from A to A_e when neither Type 1 nor 2 exists, Type 1 only exists, Type 2 only exists, and both Type 1 and 2 exist, respectively. The parameter values of $(m_1, k_1, k_2, r_1, r_2, -\Delta_r G_1^\circ$ and $-\Delta_r G_2^\circ)$ are (0.1, 10⁻², 10^{-2.5}, 10^{-0.8}, 0.5, 50 and 10) for (a), (0.1, 10⁻², 10^{-2.5}, 10^{-0.8}, 0.5, 2000 and 1600) for (b) and (0.4, 10^{-1.6}, 10^{-2.4}, 10^{-0.9}, 0.5, 50 and 10) for (c). Other parameters are set to the default values presented in Table S1.”

We also reduced the description of the figure in the main text and added some labels to Fig. 3 (Please see lines 224, 228 – 230, and 244 and Fig. 3). Hope that the revised version is easier to read.

Reply to comments from reviewer 2:

> This manuscript develops a model on a catabolic cycle between two types of microbes and > analyses its effect on their population dynamics from the perspective of thermodynamics.
> This study shows that their biomass, efficiency of resource utilization, and ecological niches > are improved by the symbiotic interaction. Based on these results, this study argues that the > first two results are caused by the cyclic structure of the catabolic interaction, while the last one > is derived from stoichiometric effects of the

> interaction. Also, this study provides insights into what structure of catabolic interaction can be possible in ecological community from thermodynamical perspective.
> Although their results that the symbiotic interaction can improve the biomass and ecological niches of microbes are not so surprising, I thought that their above discussions are very insightful, and they can provide a new perspective on studies of ecological community. Such discussion indicates that their rigorous modelling beyond intuition is worthwhile. I also thought that the model is appropriate and mostly described.
> My following comments are all fairly minor:

[Answer] We are delighted to receive such a positive evaluation. We have revised the paper in response to the comments by the reviewer.

[R2-1] > Introduction: One of the famous hypotheses of the origin of eukaryotes and mitochondria is based on a catabolic cycle (syntrophic hypothesis: Moreira & López-García (1998) *J. Mol. Evol.* 47, 517–530). This might highlight the importance of this study from not only ecological but also evolutionary aspects.

[Answer] We are very thankful to the reviewer for giving information of an important literature. We cited it in the introduction and explained the implication. (lines 92 – 95)

[R2-2] > L 78: The gas constant, R , should have the dimension of T^{-1} .

[Answer] Done in line 83.

[R2-3] > L 124 & 128: For simplification, it is assumed that A is completely recycled and concentrations of all other compounds are constant. I felt such assumptions are suitable as a first step of modelling, while it would be good if the authors touched their evaluation how results are affected by relaxing the assumptions.

[Answer] In lines 367 – 372, we introduced the preliminary results when we numerically simulated the case where the concentration of other compounds also changes with the progress of reactions:

“We consider the situation in which the concentration of B , B_e , C , and C_e are unchanged because of the presence of buffering effects. We performed a preliminary examination for the case where B , B_e , C , and C_e also changes with the progress of reactions and observed, interestingly, the oscillatory fluctuation instead of predictable and steady state behaviour reported in this paper, which will be discussed in future publication.”

[R2-4] > Eq 4a: I think there are some missing parentheses here?

[Answer] Corrected in Eq. (4a).

[R2-5] > L 140: If it is possible, it would be nice if there is an intuitive explanation of the Monod equation or the multiplicative form of Michaelis–Menten equation.

[Answer] There are several proposed structured models of substrate-utilization kinetics that encompasses dual-limitation from the abundance of electron donor and acceptor. Here we adopt the simplest model, double-Monod model. The multiplicative model assumes that, if two essential substrates are present at subsaturating concentrations, both directly limit the overall reaction rate, and the limitation effect are multiplicative. In lines 152 – 154, we added the explanation for this and cited one literature.

[R2-6] > L 272: I felt the explanation that there is no conflict between the two species as they are mutualist is not appropriate. The reason is that they do not compete for the same resources,

>isn't it?

[Answer] Agreed. In lines 283 – 284, we amended the corresponding sentence as follows:

“there is no conflict between types 1 and 2 as they do not compete for the same resource.”

[R2-7] > *Fig. 4: Although I expected the case where both species cannot invade and exist without each other, such mutual dependency seems to not occur. This model does not have multistability (L 184). Is it hard to occur in principle, or is it possible by relaxing some of the assumptions? Since such a situation highlights the importance of catabolic interactions, if the authors have any idea about it, it would be nice if the difficulty or possibility of such mutual dependency is touched.*

[Answer] We agree with the reviewer. This is an interesting question. In fact, we have observed multi-stability in a model similar to the model presented in this manuscript. However, the analysis is not completed yet and also these aspects are not the main focus of the current paper, the latter focusing on demonstrating the mutualistic catabolic relationship in bacteria consortium engaging in recycling interaction. As we will write a separate paper on these topics, we would not like to reveal these ideas that will comprise the key concepts of the new paper.

[R2-8] > *L 391: Fig. 5© → Fig. 5C*

[Answer] Done in line 428.

[R2-9] > *L 398: reaction 3 → reaction 2?*

[Answer] This should be reaction 2. To make the intention of this sentence clearer, in lines 434 – 436, we amended the corresponding sentence as follows:

“..., the realised niche of type 2 can be expanded because it can indirectly depend on reaction 3, the overall reaction of reaction 1 and 2, owing to the high productivity of reaction 1 that fuels reaction 2.

[R2-10] > *L 409: This argument is interesting.*

[Answer] Thank you for the positive feedback.

Hope the revised version is clearer than the previous version.